# Research on Tensile Properties of Carbon Fiber Composite Laminates

**DOI:** 10.3390/polym14122318

**Published:** 2022-06-08

**Authors:** Jiayi Wang, Lifeng Chen, Wei Shen, Lvtao Zhu

**Affiliations:** 1College of Textile Science and Engineering (International Institute of Silk), Zhejiang Sci-Tech University, Hangzhou 310018, China; wjoy18867578703@163.com; 2Shaoxing Baojing Composite Materials Co., Ltd., Shaoxing 312000, China; vincentchenli@sina.com (L.C.); shenw@jinggonggroup.com (W.S.); 3Shaoxing-Keqiao Institute, Zhejiang Sci-Tech University, Shaoxing 312000, China

**Keywords:** carbon fiber composite, tensile strength, laminate, tensile property

## Abstract

In order to study the thread tensile performance of carbon fiber composite laminates, the connection between the test piece, connecting bolts, bushings, and the composite matrix, was leveraged for loading, and combined with an ultra-sound scanning imaging system, experiments were carried out on the dynamic response to record the failure behavior of the laminate structure of equal thickness. The effects of different pull-off loading strengths on the dynamic failure process, deformation profile, midpoint deformation, failure mode, and energy dissipation ratio of the thread were studied. The results show that (1) with the increase in pull-off strength, the response speed of mid-point deformation increases, the thread deformation mode changes from overall deformation to partial deformation, and the localized effect increases, accompanied by severe matrix and fiber fracture failure; (2) the thread energy dissipation ratio ascends with increasing pull-off strength and exhibits three distinct stages, i.e., elastic deformation, central fracture, and complete failure, which are directly related to the structural failure mode; (3) the failure load increases with the increment of the thickness of the laminate, and the maximum failure surface of the specimen will move from the upper layer of the laminate to the lower layer along the thickness direction; (4) the deformation velocity of the midpoint augments with the increase in the tensile rate, which can be included as a factor to assess the tensile properties of carbon fiber composites.

## 1. Introduction

Carbon fiber reinforced polymer (CERP), with its high specific strength, tailored modulus, and sound stealth absorbing performance, has replaced some traditional metal materials and material structures. As an indispensable assembly of the modern aviation industry [1], carbon fiber and its composite materials represent significant advantages with regard to fatigue resistance, tensile resistance, vibration damping, volatile temperature resistance, and corrosion resistance. However, the composite structures also have some defects and are easily damaged under long-term loading, machining, and corrosion due to factors such as production process, design scheme, and retention environment. In order to effectively prevent secondary accidents, including internal and external structural damage, it is necessary to carry out real-time quality and safety inspections when they are put into use to ensure mechanical integrity and production safety.

The early research work on the dynamic failure behavior of fiber reinforced composite laminates is mainly based on the contact tensile loading formed by the drop weight impact and the penetration of the projectile. The research [2] results show that the main failure modes of fiber reinforced composite laminates include matrix and fiber fracture, spalling, and so on. Schiffer et al. [3] used an underwater tensile loading simulation device to explore the response of laminates under high-strength underwater tensile loads and established a theoretical analysis model for the dynamic response of composite laminates. Yang et al. [4] leveraged three-dimensional DIC to mine the transverse dynamic response process of carbon fiber woven laminates under projectile penetration.

Rajput et al. [5] studied the impact of laminate thickness on the impact response and damage mechanism through experiments and numerical analysis, and the results showed that the depth of pits had a bilinear response to the strength exerted. Karalis, G [6] studied the dynamic behavior and failure mechanism of laminates under underwater tensile loads. Lin et al. [7] conducted a comparative analysis of the dynamic failure behavior of flat and curved panel structures of woven basalt/epoxy laminates under blast loading, emphasizing the important influence of structural curvature on the dynamic response. Wei et al. [8] carried out experiments and numerical simulations on the high-speed penetration of laminates and concluded that, in addition to fiber failure, laminates under penetration loads also include spalling and matrix cracking. Penetration velocity, penetration angle, fiber layup, and laminate structure form generate rich research results on the dynamic behavior and failure of laminate structures under tensile loads [9]. Based on the Hashin failure criterion, Wang Danyong [10] and Su Rui [11] established a progressive damage analysis model for composite materials with similar ideas and also provided a life calculation result that was close to the experiment. Saeedifar et al. and Yang [12,13] established the finite element model of interlaminar fracture toughness test of fiber reinforced composites based on integral and VCCT technology, respectively, to study the delamination processes, such as crack generation and crack propagation, and the energy release rate during interlaminar failure. Three-point bending tests with complex failure forms are rarely studied, and the simulation is not accurate. Camanho [14] proposed that the initial damage inside the composite structure was mainly caused by the separation between fiber layers. Combining with the maximum stress theory, and based on the three-dimensional progressive failure constitutive model, Pearce [15] explored the stiffness degradation characteristics of composite connection structures under out-of-plane loads.

Pravalika and Kashi Ishikawa [16] studied the strength of composite joints through experiments and divided the failure types of carbon fiber composite into four types, namely fiber failure, matrix failure, interlaminar delamination, and normal shear failure, using ultrasonic scanning imaging system and observing the internal damage of the composite material. Based on the observation results, the damage process of the specimen was divided into four stages, namely the appearance of damage, the extension of local cracks, the expansion of damage, and the appearance of structural cracks. Ostapiuk and Orifici [17,18] studied the effect of the ratio of aperture to plate thickness on the connection performance. The results show that when the ratio of aperture to plate thickness is equal to 1, the bearing capacity of the composite plate is the strongest. MA Mc Patel [19] and Whitworth [20] analyzed the stress of single-nail and multi-nail connections by means of tests. The test results show that as the number of nail holes in the laminate increases, the strength of the laminate decreases. Ang [21] combined the failure criterion and the maximum stress criterion proposed by Hashin to study the mechanical connection characteristics of composite materials through experimental research and numerical simulation and predicted the strength and failure process. Zhou [22] and Zhang [23] pointed out that the pull-off failure mode is the same as the internal failure characteristics caused by low-speed stretching, mainly manifested as matrix failure and delamination; the damage extends from the edge of the hole to the surrounding area, and the damage area is distributed in a network along the thickness direction. 

It can be derived from the above statement that the process of pull-off failure is very complex, and there are many factors affecting the pull-off strength of laminates, such as temperature and humidity conditions, ply ratio, material geometric parameters and constraints, etc. Due to many other reasons, there are relatively few studies on the pull-off characteristics of composite materials, and there are still many problems that need to be further explored.

This work aims at the dynamic response and failure mode of the carbon fiber composite laminates under the tensile load generated in the thread pull-off. Along with the ultrasonic characteristic scanning imaging system, the damaged specimens are tested. The pull-off test of composite laminates is carried out to obtain the pull-off strength; the load–displacement curve of the entire pull-off behavior is recorded; the layer-based damage are studied; the failure processes and the mechanical response of the laminate are analyzed. The numerical simulation results are to be further explored, focusing on the main failure modes of the laminates, i.e., the initiation and expansion of damage in the laminate layers, and their effects on the pull-off strength of the composite laminates. 

## 2. Experiments

### 2.1. Experimental Materials

The carbon fiber composite laminate used in this study is the T700 carbon fiber composite single-layer board. The material properties of the T700 carbon fiber composite single-layer panels are as follows: longitudinal stiffness $E_{1}=100\;\mathrm{GPa}$ transverse stiffness $E_{2}=80\;\mathrm{GPa}$, Poisson’s ratio $\nu_{12}=0.21$, shear modulus $G_{12}=4\;\mathrm{GPa}$, longitudinal tensile strength $X_{T}=2100\;\mathrm{MPa}$, longitudinal compressive strength $X_{C}=700\;\mathrm{MPa}$, transverse tensile strength $Y_{T}=42\;\mathrm{MPa}$, transverse compressive strength $Y_{C}=160\;\mathrm{MPa}$, interlaminar shear strength S=104 MPa, density $\rho =1500\;\mathrm{kg}/m^{3}$.

The thread pull-off (rear joint) used in this study adopts high-temperature T700 spreading cloth to form the RS03A composite material rear joint according to the requirements specified in the task book. The rear joint of the RS03A composite material is selected for the thread pull-off test. The composite material layer (matrix) and the bushing are composed of TC4 material and equipped with a boss at the bottom to bear the axial load of the bushing. The measured axial tensile properties and stress–strain curves are shown in Figure 1 and Table 1. 

Figure 1 illustrates that under the action of the axial tensile load, the stress and strain of the threaded specimen result in a smooth linear relationship. No fracture signs occur in the figure. According to the measured stress–strain curve, the axial tensile fracture strain is about 0.9%. It can be seen from Table 1 that the average axial tensile fracture strength of the thread is 1565 MPa, and the average modulus is 170.5 GPa.

### 2.2. Experiment Equipment

The test piece and the adapter are connected and tightened by bolts; the adapter is connected with the slider, and then, the adapter is inserted into the indenter. The above-mentioned overall structures are placed on the inclined blank component fixed on the load-bearing ground rail, and the actuator is connected with the slider through a multi-strand wire rope, as shown in Figure 2. The loading control equipment for the pull-off and bending test of the laminate test piece adopt a multi-channel coordinated loading control system. The error of the coordinated loading control system is less than 1%, which met the requirements of the task book for loading accuracy.

### 2.3. Experimental Installment

During the experiment, the INSTRON1346 electro-hydraulic servo-controlled material testing machine was used to carry out tensile experiments and laminate compression experiments, as shown in Figure 2 and Figure 3. The data acquired in real time during the loading process of the test system include load, displacement, and transverse deflection. Two billets are installed at the appropriate position below the actuator with anchor bolts. The combination of the pull-head adapter and the test part is placed in the pressure head and placed in the center.

## 3. Process

### 3.1. Test Process

Before starting the test, a static load spectrum is created. The loading system and displacement (test parts are required to be close to the fixture) are set to zero. Different load intensities are exerted to monitor the displacement. During the tensile process, the initial failure load, maximum load, and corresponding displacement are recorded with displacement control until the specimen is fractured. 

### 3.2. Test Status

The ultimate tensile strength is the maximum tensile stress that carbon fiber composites can withstand before reaching failure under the tensile test load. 

Tensile strength calculation formula
(1)$$\sigma_{t}=\frac{P_{\max}}{bh}$$

Tensile modulus calculation formula
(2)$$E_{1}=\frac{\Delta Pl}{bh\Delta l}$$

In the formula, the ultimate tensile strength is $\sigma_{t}$, the maximum load $P_{\max}$, the specimen width b, the specimen thickness h, the tensile modulus $E_{t}$. The maximum of the displacement load of the tensile test is shown in Figure 4. The test data of each group of test pieces are calculated to obtain an average tensile strength of the laminate of 783.23 MPa, an average limit load of 41.97 KN, and an average Poisson’s ratio of 0.317, as shown in Table 2. The tensile strength, tensile elastic modulus, and Poisson’s ratio data obtained from the sample are similar, which shows that the heterogeneity of the carbon fiber composite sample is within the experimental error range. The obtained test results are well performed. 

According to the literature [24,25], the bending strength can be expressed as
(3)$$\sigma_{f}=3PL/(2bh^{2})$$

In the formula: $\sigma_{f}$ is the bending strength, MPa; P is the maximum load value when the sample fails, N; L is the span, mm; h is the thickness of the sample, mm. 

The flexural modulus of elasticity is
(4)$$E_{f}=\Delta PL^{3}/(4bh^{3}\Delta f)$$

Shown in the Equation (4) is the calculation of the flexural modulus of elasticity. $E_{f}$ is the flexural modulus of elasticity, MPa; ΔP is the load increment of the initial straight-line segment on the load-deflection curve, N; $\Delta_{f}$ is the corresponding deflection increment, mm, at the midpoint of the sample span. The test results are shown in Table 3.

### 3.3. Finite Element Model

Three-dimensional finite model is constructed to analyze the laminate, as shown in Figure 5. The element type adopts the eight-node reduction integral solid element C3D8R, and the orientation of lamination is realized through material orientation. When the load is applied, the left end of the model is fixed, and the right end is applied with axial displacement load. Different strain rate conditions are realized by adjusting the step length of the analysis. The time domain is set as 0.05 to simulate the quasi-static loading mode of force in the pull test.

### 3.4. Damage Monitoring

As the composite laminate is damaged by pulling, the failure mode of the hole edge is fuzzy, and the vidual defects are not very distinct from each other. Therefore, it needs to be observed by non-destructive testing equipment. In this test, the ultrasonic characteristic scanning imaging system (UTF-SCAN-1 Water immersion C-scan detection system) performs non-destructive testing on the damaged specimens. The location, size, and damage plan of the damaged area of the laminate can be obtained by ultrasonic C-scanning. The C-scan non-destructive testing test is carried out on the test pieces of different thicknesses after the pull-off test, and the damage of each layer of the test pieces with different thicknesses, area, and depth is obtained. As shown in Figure 6a–c above, the maximum failure surface of the specimen with a thickness of 1.25 mm is located in the lower layer of the laminate, that is, the fourth layer (90-degree direction), and the specimen with a thickness of 3.25 mm has the largest failure surface, while the failure surface is located in the middle layer of the laminate, that is, the ninth layer (90-degree direction). The maximum failure surface of the specimen with a thickness of 5.10 mm is located in the upper layer of the laminate, that is, the seventh layer (0-degree direction). It can be seen that with the increase in the thickness of the specimen, the position of the maximum failure surface of the specimen moves along the thickness direction from the upper layer position (including the straight hole surface) of the laminate to the lower layer position (including the countersunk hole surface). 

## 4. Results and Discussion

### 4.1. Analysis of Tensile Properties of Carbon Fiber Composites

After, respectively, exploring the axial tensile properties of composite materials and threaded composite materials, the rear joint of RS03A composite material is selected for thread tensile testing. The load–displacement curve during the tensile process is shown in Figure 7. As can be seen from Figure 7, when the load is applied to a certain level (point A in Figure 7), it will suddenly drop, and the composite material will emit a crisp sound during the experiment; the load at this time is defined as the initial failure load [26]. As the displacement increases, the load continues to increment, and the sound continues to appear during the period until the last loud sound. The cylinder composite material breaks as a whole, and the maximum load at this time is the breaking load. The initial failure load and fracture load are used to calculate the stresses, which are recorded as the initial failure stress ($\sigma_{1}$) and the final failure stress ($\sigma_{2}$).

At the beginning of the experiment, the appearance of the laminate basically does not change, and the stress–strain curve shows a linear rise. When the strain reaches 4500 με, the edge of the hole in contact between the laminate and the adapter begins to have a small fiber uplift, and there is a clear and crisp squeak, indicating that the laminate begins to damage. As the displacement continues to increase, the area of the uplifted fiber increases slowly, and the damage range expands from the center of the circular hole to the edge of the laminate, and some fibers are pulled off at the edge of the hole. The load drops slightly, the curve of the increase in stress continues to rise slowly. As the load continues to drop, it is accompanied by a continuous brittle sound. Finally, the “explosion” is carried out in the “middle section” of the failure mode until the laminated plate is completely destroyed. Fiber fracture, delamination, fiber pull, and debonding are found at all locations, indicating that the matrix and interface are severely damaged during loading.

The stress–strain curve of axial tension is shown in Figure 8. It can be seen that the axial tensile modulus of the composite samples is relatively stable, with an average value of about 90 GPa. The average initial failure stress is about 470 MPa, and the final failure stress average is about 800 MPa. Shown in Figure 9 is the photo of the composite material after fracture. It can be seen from the figure that when the composite material is finally damaged, the fiber is fractured. At the same time, it can be seen that the fiber layer also cracks many times during the fracture. The final breaking load of the composite material is caused by the fracture of the helical fiber as the main force carrier. When the first load occurs, the hoop layer composite material has already cracked. At this time, its initial failure stress is about 470 MPa, and the axial tensile fracture strain is about 0.5%, manifesting that the decrease in the first load force in value is due to the failure of the composite laminate reaching the breaking strain. In the actual working conditions of the composite material used in this experiment, if cracking occurs, it signifies a functional failure. Therefore, the initial failure stress when the load decreases for the first time is used as the criterion for judging whether the composite material fails.

The picture after fracture of composite material is shown in Figure 9. It can be seen from the picture that fiber fracture occurs when the composite material is finally destroyed. At the same time, it can be seen that the fiber layer of the fracture also occurs several times. The final fracture load of the composite is large, which is caused by the fracture of the helical fiber. The test curve shows that the tensile failure of composite laminates is divided into two stages, and it is a nonlinear and progressive failure process.

### 4.2. Layer-Based Damage

Figure 10a,b shows the interfacial delamination damage at 5–100% loading percentage, respectively. The matrix cracking first appears on the back of the laminate, the damage occurs in the middle area of the laminate, and the damage area of the unit layer farther from the tensile side is larger than that near the tensile side, which can be explained by the deformation and failure principle of the laminate, that is, matrix tensile damage starts from the back and extends to the upper layer [27]. With the increase in tensile energy, the cracked area of the matrix gradually expands. The damage profile of each layer is roughly an irregular ellipse and expands along the fiber direction. This is because the stress is transmitted faster in the fiber direction, so the damage profile in the fiber direction is larger, which is consistent with the experimental results.

The figures indicate that delamination occurs at each interface with varying degrees of damage. When the damage variable is equal to 1, complete delamination is indicated. The main axis of the delamination area (45°/−45°) is along the −45° direction, and the main axis of the delamination area (−45°/45°) is along the 45° direction, that is, the main direction of the delamination damage is along the laying of the fibers close to the direction of the lower layer. It can also be seen that, regardless of the load percentage, the interface near the back of the tensile point is the first to experience delamination damage with the largest damage area. However, the damage variables of the other layers are between 0 and 1, which only achieves a partial delamination effect. With the same energy, the closer the sublayer to the tensile side, the smaller the degree of delamination damage. This is because when the laminate is stretched, the sublayers farther from the stretched side are subjected to greater tensile stress than the sublayers adjacent to the stretched side, so the damage propagates from the bottom layer to the top layer. Additionally, with the increase in energy, except for the bottom layer, the delamination damage area of other layers also becomes larger.

### 4.3. Based on Mechanical Response

Under the pressure of out-of-plane load, the mechanical response of the pull-out failure process of the connected structures demonstrates the following four stages.

The bolt stays in the pre-tightening stage, and the load is transferred through the contact pair of the model. The materials of each component are in the elastic stage, and the connecting structure experiences no macro damage. In the model, the damage variable value of the cohesive force unit increases from 0, and there is microcrack damage between material interfaces. At this stage, during the initial loading process, the load and displacement increase rapidly until the maximum load is reached.

During the plastic stage, the stiffness of the connecting structure decreases [28]. Weak plastic deformation occurs at both ends of the screw hole, the increasing rate of load slows down, and the contact pair between the bolt and the screw hole produces weak slip. At this stage, the displacement of the laminated plate continues to increase, but the bearing capacity fluctuates up and down, indicating that the matrix of the laminated plate had been damaged and can no longer bear the load, and internal damage had occurred, leading to the decline of stiffness.

The carbon fiber composite has anisotropic characteristics, and the pulling load is in the direction of the laminate method, with bearing performance [29]. When the load displacement is increased to about 0.58 mm, the damage variable value in the model reaches the failure limit, and the fiber layer is separated in the area near the screw hole, resulting in tension, compression, and shear failure of fiber and matrix material. When the stress of the cohesive element reaches the fracture toughness value of the interface, the adapter disconnects from the fiber layer. The internal micro-clearance of laminates and partial contact failure at the screw hole lead to the deterioration of the stiffness of the connecting structure. Figure 11 shows the fiber damage status after tensile failure.

Conical uplift occurs near the hole of the laminates. Due to the extrusion of the bolt contact surface, the micro-gap inside the composite is compressed, the stiffness of the connection structure is appropriately increased, and the slope of the response curve is slightly increased. When the load reaches the ultimate stress value, the metal in the ring region of the head produces crushing failure [30]. The ultimate load is the tensile strength of the connecting structure.

### 4.4. Failure Mode Based

Due to the continuous loading of the equipment, the laminate will slowly deform and fail as the loading strength of the jack increases. The central loading area of the laminate beam is subject to the maximum stress due to bending/tensioning [31,32]. Being exposed to the action of incident compressive stress waves and bending waves, the threads will form lateral deformations and localized wrinkles at the edges in contact with the laminate beams. The compressive stress wave is reflected by the backside of the laminate to form a tensile wave. When the tensile wave intensity is large enough, the laminate will undergo spalling between the fiber and the matrix. Under sufficient loading strength, with the increase in transverse deformation and axial tensile, the fracture failure of the matrix and fibers occurs in the laminate.

Figure 12 compares the deformation profile with time and strength at 50% and 100%, showing the obvious localization of deformation. When the loading strength is between 5% and 50%, and the velocity is 1.5 mm/min, the two sides of the laminate always slip along the radial direction without lateral deformation. However, when the velocity is set at 1 mm/min, the transverse defection will move sharply. When the loading percentage is 40–50%, the deformation gradually diminishes. With the increment of structural deformation, the target plate finally leaves the fixture and continues to move with a certain kinetic energy. When the loading percentage is greater than 50%, the laminate mainly undergoes elastic deformation, and no obvious failure occurs on the surface of the laminate. When the loading percentage reaches 60%, the laminate fails with the increase in tensile strength. The laminate is completely broken when the load reaches 90%.

## 5. Conclusions

An experimental study on tensile properties of carbon fiber composite laminates was carried out.The damage condition of the laminates’ layer, the mechanical response, and failure mode were discussed.

(1)It can be concluded from the load–displacement curve of the pull-off that the pull-off failure of the composite laminate is nonlinear and conforms to the principle of progressive damage. With the pressure of out-of-plane load, the mechanical response of the screw structure presents a nonlinear trend and exhibits the process in four stages. The characteristics of pull-out failure are similar to impact failure. The conical uplift of the connecting hole area and the separation between the layers of the fiber–metal interface are the main factors leading to pull-out failure.(2)As the tensile load increases, the damage area of various damage types also becomes larger. When the tensile energy increases to a certain degree, the load will drop twice, and the damage will occur twice. In this paper, for the first time, when the load increased from the initial value to 55 KN, the contact edge between the laminate and the pull rod is raised, resulting in large area damage. At the same time, the load decreases, which is the initial failure. The second time is when the load reaches 65 KN, the fiber at the hole edge of the laminates is pulled off, and the damaged area expands from the hole edge to the surrounding area. The failure part is mainly concentrated near the edge of the hole on the surface of the laminate (including the straight hole surface), and the fiber at the failure witnesses a whole piece of bulge.(3)With the increase in pull-off strength, the failure of carbon fiber composite panels is mainly divided into micro-deformation under low-load pull-off, half-fold fracture under medium-load strength impact, and complete fracture under high-strength load tensile, and exhibits structural failure modes. The failure mode of the composite laminate is a non-fracture failure of local nature, with high safety.(4)The deformation velocity of the midpoint increases with the increase in the tensile rate. When the rate and load increase simultaneously, localization failure occurs, and the critical maximum deformation of laminates decreases with the increase in the rate. Both the loading strength and tensile rate can be used as factors to assess the tensile properties of carbon fiber composites.

## Figures and Tables

**Figure 1 polymers-14-02318-f001:**
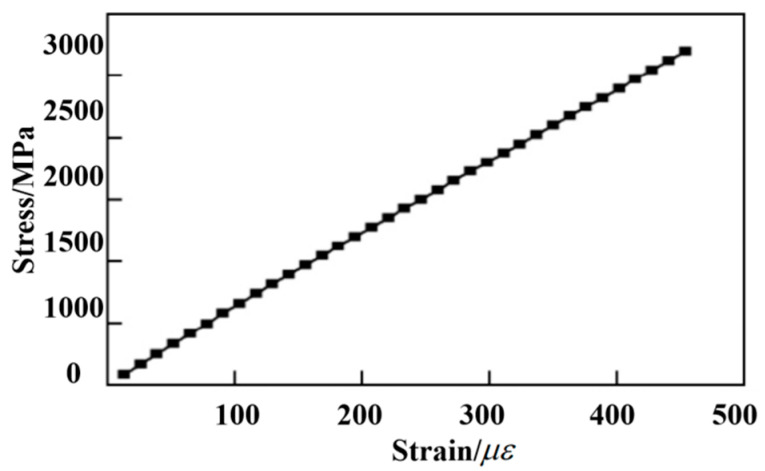
Thread axial tensile stress–strain curve.

**Figure 2 polymers-14-02318-f002:**
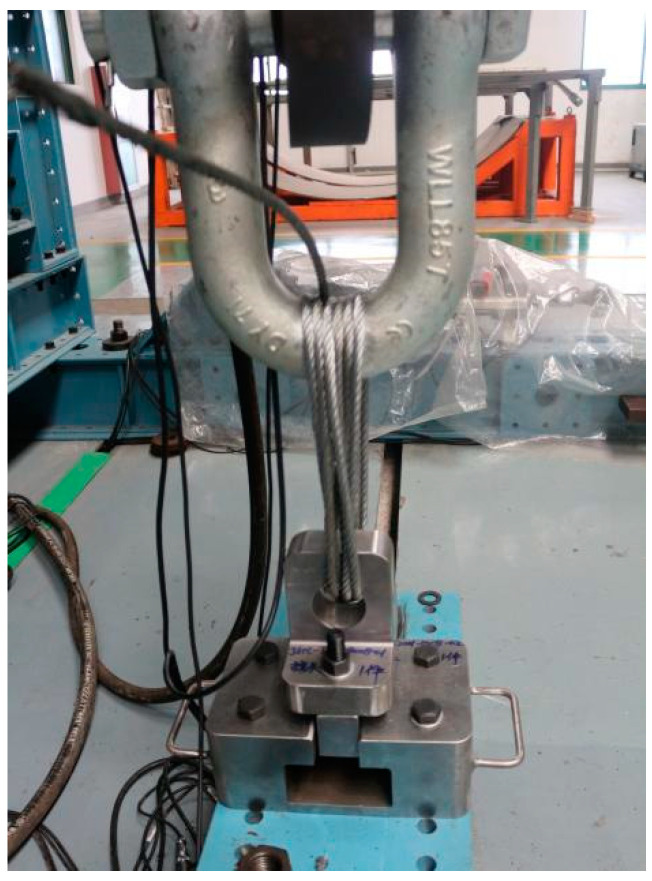
Overall condition after pull-off test installation.

**Figure 3 polymers-14-02318-f003:**
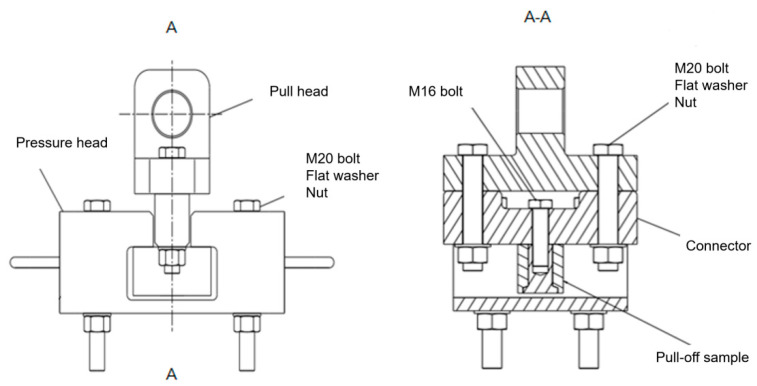
Schematic diagram of the installation of the thread pull-off test piece.

**Figure 4 polymers-14-02318-f004:**
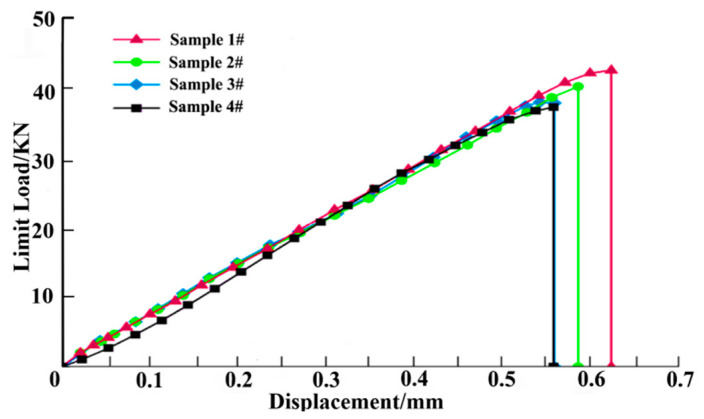
Limit load–displacement curve.

**Figure 5 polymers-14-02318-f005:**
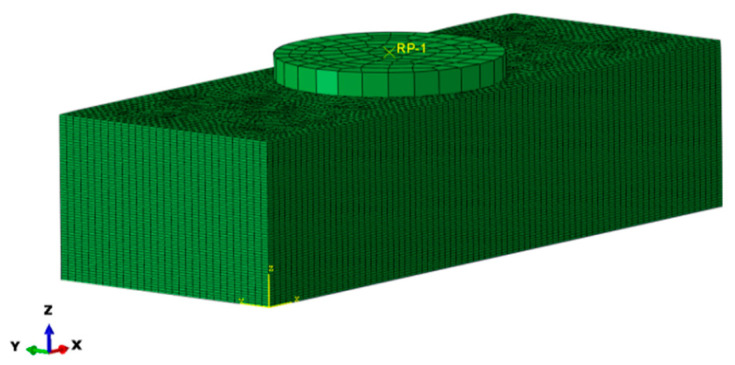
Finite element model of composite laminate assembly.

**Figure 6 polymers-14-02318-f006:**
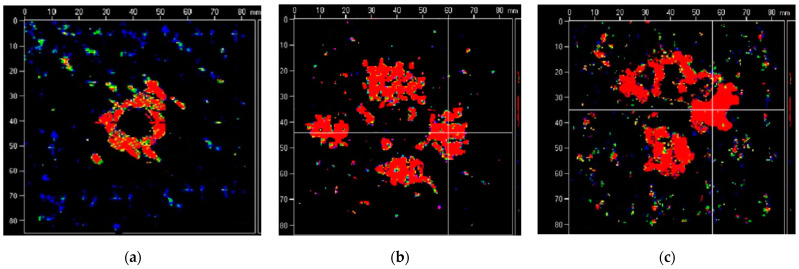
The maximum failure surface and depth of the specimen with different thickness: (**a**) the fourth layer, 1.25 mm, (**b**) the ninth layer, 3.25 mm, (**c**) the seventh layer, 5.10 mm.

**Figure 7 polymers-14-02318-f007:**
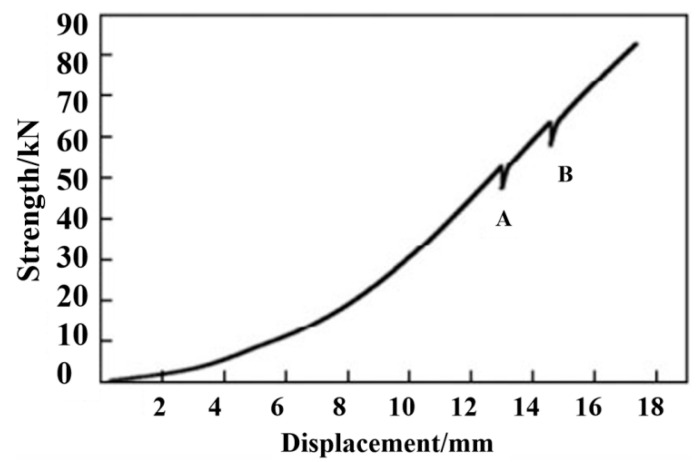
Load–displacement curve of axial tension.

**Figure 8 polymers-14-02318-f008:**
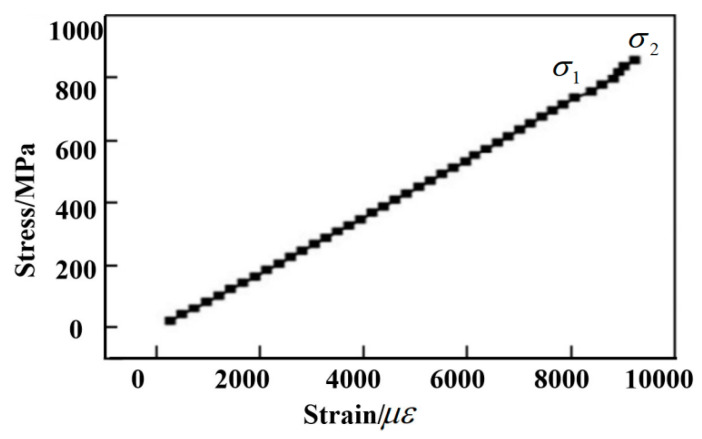
Stress–strain curve of axial tension.

**Figure 9 polymers-14-02318-f009:**
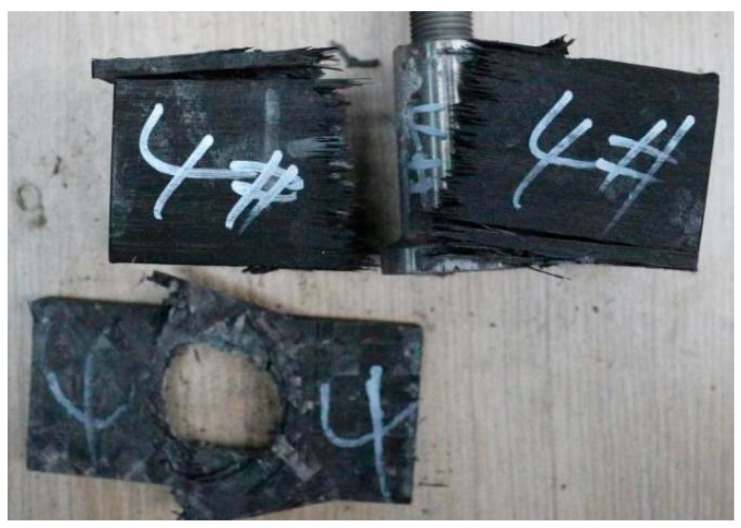
Fracture photo.

**Figure 10 polymers-14-02318-f010:**
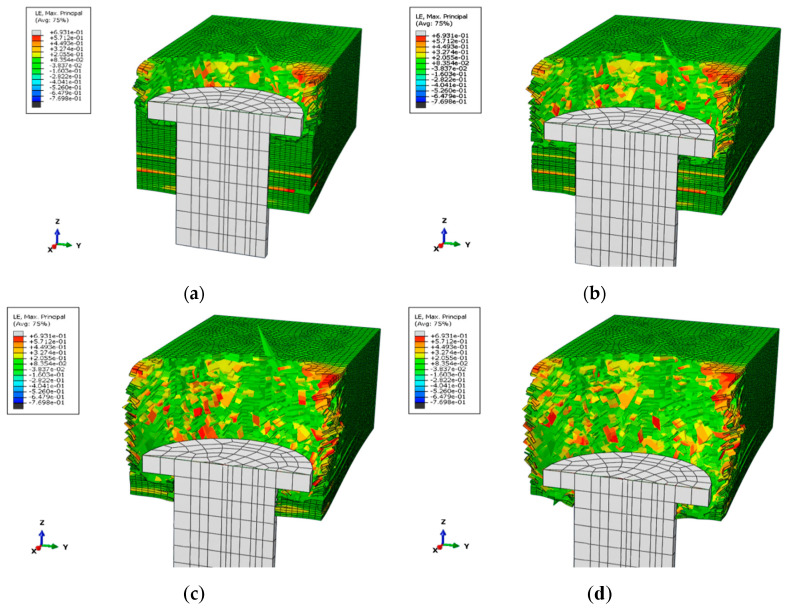
Matrix tensile damage distribution under load of different percentage: (**a**) load intensity 5 × 10^−2^ to 25 × 10^−2^, (**b**) load intensity 25 × 10^−2^ to 50 × 10^−^^2^, (**c**) load intensity 50 × 10^−2^ to 75 × 10^−2^, (**d**) load intensity 75 × 10^−2^ to 100 × 10^−2^.

**Figure 11 polymers-14-02318-f011:**
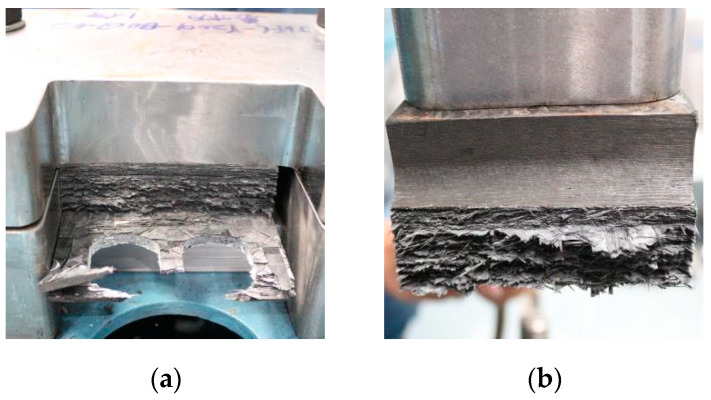
Fiber damage status (**a**,**b**).

**Figure 12 polymers-14-02318-f012:**
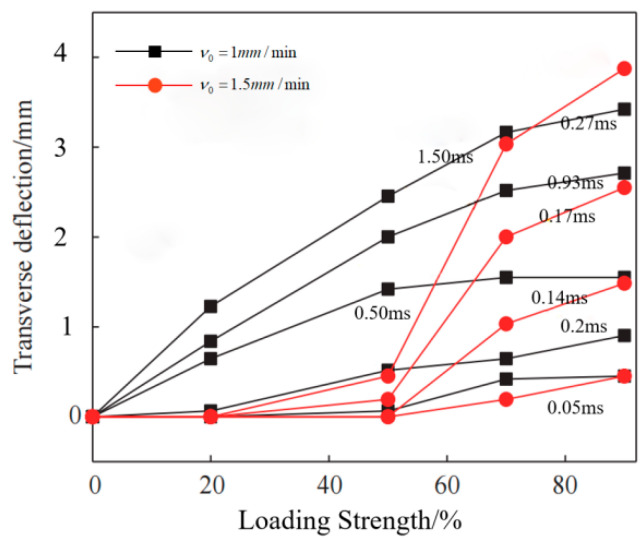
Deformation profiles of laminates at different loading strengths and velocities.

**Table 1 polymers-14-02318-t001:** Axial tensile properties of threads.

Sample	Module/GPa	Tensile/MPa
1#	165.6	1615
2#	178.8	1497
3#	166.8	1544
4#	170.9	1604
Avg. value	170.5	1565
Dispersion coefficient	3.50	3.52

Note: “#”stands for nothing here. It is a symbol marked on the sample part to tell one digit apart from another.

**Table 2 polymers-14-02318-t002:** Laminate tensile test results data.

Sample	Compressive Strength (MPa)	Limit Load (KN)
1#	830.43	43.01
2#	884.90	43.59
3#	743.46	41.21
4#	690.16	40.08
Average	787.23	41.97

Note: “#”stands for nothing here. It is a symbol marked on the sample part to tell one digit apart from another.

**Table 3 polymers-14-02318-t003:** Test results.

Sample	Bending Strength/MPa	Flexural Modulus of Elasticity/GPa
1#	1068	162
2#	1053	137
3#	995	155
4#	989	131

## Data Availability

Data sharing not applicable.

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
