# Peer review of "Research on Tensile Properties of Carbon Fiber Composite Laminates"

_polymers, 2022, doi:10.3390/polym14122318_

Round 1
Reviewer 1 Report
In the Reviewer opinion the research paper entitled “Research on Tensile Properties of Carbon Fiber Composite Laminates” is average.
This research provides a theoretical basis for the design of composite shell joints. The effects of different pull-off loading strengths on the dynamic failure process, deformation profile, midpoint deformation, failure mode and energy dissipation ratio of the thread were studied. The results show that with the increase of pull-off strength, the response speed of mid-point deformation increases, the thread deformation mode changes from overall deformation to partial deformation, and the localized effect increases, accompanied by severe matrix and fiber fracture failure
Some comments which greatly enhance the understanding of the paper and its value are presented below. Specific issues that require further consideration are:
- The title of the manuscript is matched to its content.
- The structure of the manuscript is rather proper.
- The Introduction covers the cases.
- In the Reviewer’s opinion, the current state of knowledge relating to the manuscript topic has not been covered and clearly presented.
- An analysis of the manuscript content and the References shows that the manuscript under review constitutes a summary of the Author(s) achievements in the field.
- Article has flaws, additional experiments needed, research not conducted correctly.
- In the Reviewer’s opinion, the bibliography, comprising 23 references, is not representative and exhaustive.
- I suggest expanding the conclusions.
- In the Reviewer’s opinion the manuscript can be published in the journal, but after major revision.
Reviewer 2 Report
The manuscript “Research on Tensile Properties of Carbon Fiber Composite Laminates” by J. Wang et al. is dedicated to the experimental study of the thread tensile performance of carbon fiber composite laminates. Such studies are relevant to a certain extent due to the widespread use of composite materials based on carbon fibers and the presented work could be of partial interest to readers of the “Polymers” specializing in the field of synthesis and characterization of polymer-based composites with a fibrillar structure. However, the way in which the results are presented is subject to serious criticism, and I cannot recommend the manuscript for publication in its present form. The corresponding comments and questions are listed below.
- The quality of English writing is sometimes bad and not free of typos; I recommend that the authors make a thorough check of the text and make it clearer for reading.
- The manuscript is replete with secondary and often unnecessary details of the experiment (in particular, the dimensions of the bolts used, their tightening torques, etc.). On the other hand, the manuscript lacks any detailed analysis of the errors in the results obtained. As a particular example, Tables 2 and 3 present the results of mechanical parameter estimates, including from 4 to 5 significant digits. Does this mean that these estimates were made with errors significantly less than 1%? In any case, the confidence intervals should be presented for all the experimentally measured data (together with the error bars on curves).
- That is mu-epsilon in the X-axis notation (Fig. 1)?
- Line 64, “…using SEM (Standard Electronic Components)….” - ? Maybe, the Scanning Electron Microscopy?
- Line 236, “….and the composite material will emit a crisp sound…”. The acoustic emission from local damaged zones in stressed materials is a well-known effect widely used in the material science over several decades. Some additional comments regarding this item are necessary.
- The subsection 4.4. The discussion regarding Fig. 11 is unclear. Was this curve obtained in the experiments? If so, the clarifying comments regarding the experimental technique and relationship with other experimental data are necessary. What are the units for the axes notations?
- The Conclusion section. At least some of the conclusions seem fairly obvious and trivial (for example, “As the tensile load increases, the damage area of various damage types also becomes larger” (line 348). Also, the concept of “doubled damage” is unclear. It seems that the damage occurrence is the final stage of any crash test. It would be better if the authors provided in the “Conclusion” section some quantitative characteristics specific to the case under study.
- It is mentioned the Abstract that “This research provides a theoretical basis for the design of composite shell joints.” Unfortunately, I did not find any quantitative theoretical basis in this work, except for qualitative reasoning, at times debatable.
Reviewer 3 Report
The work concerns a very current research problem regarding the use of carbon fibers in technology. These fibers promise new possibilities and mechanical properties, but they are very difficult in the technology of producing composites. The authors have presented a very interesting experiment with these fibers. They reliably developed the results of the research, and presented the conclusions of these studies in the summary. In my opinion, this is a very valuable work.
Round 2
Reviewer 2 Report
The quality of paper has been improved after introducing the recommended revisions. I can recommend it for further publication.